# The hepcidin regulator erythroferrone is a new member of the erythropoiesis-iron-bone circuitry

**Melanie Castro-Mollo[1†], Sakshi Gera[2†], Marc Ruiz-Martinez[1], Maria Feola[1], Anisa Gumerova[2], Marina Planoutene[1], Cara Clementelli[1], Veena Sangkhae[3], Carla Casu[4], Se-Min Kim[2], Vaughn Ostland[5], Huiling Han[5], Elizabeta Nemeth[3], Robert Fleming[6], Stefano Rivella[4], Daria Lizneva[2], Tony Yuen[2], Mone Zaidi[2], Yelena Ginzburg[1]\***

[1]Division of Hematology Oncology, Tisch Cancer Institute, Icahn School of Medicine at Mount Sinai, New York, United States; [2]The Mount Sinai Bone Program, Departments of Medicine and Pharmacological Sciences, and Center for Translational Medicine and Pharmacology, Icahn School of Medicine at Mount Sinai, New York, United States; [3]Center for Iron Disorders, University of California, Los Angeles (UCLA), Los Angeles, United States; [4]Department of Pediatrics, Division of Hematology, and Penn Center for Musculoskeletal Disorders, Children's Hospital of Philadelphia (CHOP), University of Pennsylvania, Perelman School of Medicine, Philadelphia, United States; [5]Intrinsic Lifesciences, LLC, LaJolla, United States; [6]Department of Pediatrics, Saint Louis University School of Medicine, St Louis, United States

**\*For correspondence:**
yelena.ginzburg@mssm.edu

[†]These authors contributed equally to this work

## Abstract

**Background:** Erythroblast erythroferrone (ERFE) secretion inhibits hepcidin expression by sequestering several bone morphogenetic protein (BMP) family members to increase iron availability for erythropoiesis.

**Methods:** To address whether ERFE functions also in bone and whether the mechanism of ERFE action in bone involves BMPs, we utilize the $Erfe^{-/-}$ mouse model as well as β–thalassemic ($Hbb^{th3/+}$) mice with systemic loss of ERFE expression. In additional, we employ comprehensive skeletal phenotyping analyses as well as functional assays in vitro to address mechanistically the function of ERFE in bone.

**Results:** We report that ERFE expression in osteoblasts is higher compared with erythroblasts, is independent of erythropoietin, and functional in suppressing hepatocyte hepcidin expression. $Erfe^{-/-}$ mice display low–bone–mass arising from increased bone resorption despite a concomitant increase in bone formation. Consistently, $Erfe^{-/-}$ osteoblasts exhibit enhanced mineralization, $Sost$ and $Rankl$ expression, and BMP–mediated signaling ex vivo. The ERFE effect on osteoclasts is mediated through increased osteoblastic RANKL and sclerostin expression, increasing osteoclastogenesis in $Erfe^{-/-}$ mice. Importantly, $Erfe$ loss in $Hbb^{th3/+}$ mice, a disease model with increased ERFE expression, triggers profound osteoclastic bone resorption and bone loss.

**Conclusions:** Together, ERFE exerts an osteoprotective effect by modulating BMP signaling in osteoblasts, decreasing RANKL production to limit osteoclastogenesis, and prevents excessive bone loss during expanded erythropoiesis in β–thalassemia.

**Funding:** YZG acknowledges the support of the National Institute of Diabetes and Digestive and Kidney Diseases (NIDDK) (R01 DK107670 to YZG and DK095112 to RF, SR, and YZG). MZ acknowledges the support of the National Institute on Aging (U19 AG60917) and NIDDK (R01 DK113627). TY acknowledges the support of the National Institute on Aging (R01 AG71870). SR

acknowledges the support of NIDDK (R01 DK090554) and Commonwealth Universal Research Enhancement (CURE) Program Pennsylvania.

## Introduction

It has become increasingly clear that both erythropoiesis and skeletal homeostasis are susceptible to changes in iron metabolism, especially during stress or ineffective erythropoiesis. Diseases of ineffective erythropoiesis, such as β-thalassemia, one of the most common forms of inherited anemia worldwide (*Weatherall et al., 2010*), are thus associated with bone loss, primarily at cortical sites (*Haidar et al., 2011*; *Vogiatzi et al., 2006*). β-thalassemia results from β-globin gene mutations that cause ineffective erythropoiesis, splenomegaly, and anemia (*Weatherall, 1998*; *Pootrakul et al., 1988*; *Rund et al., 2005*; *Centis et al., 2000*). Patients with homozygous mutations have either red blood cell (RBC) transfusion-dependent β-thalassemia major (TDT) or a relatively milder anemia, namely non-transfusion-dependent β-thalassemia intermedia (NTDT).

Both TDT and NTDT generally present with anemia and iron overload, requiring iron chelation therapy. Surprisingly, however, TDT patients show more marked decrements in bone mineral density (BMD) compared with NTDT, despite chronic RBC transfusion that suppresses expanded and ineffective erythropoiesis. Optimization of RBC transfusion has reduced the frequency of overt bone disease, such as frontal bossing, maxillary hyperplasia, and limb deformities, and importantly, has enabled prolonged survival (*Kwiatkowski et al., 2012*). Nonetheless, growth patterns have not significantly improved (*Wallace et al., 2009*), and low bone mass remains a frequent, significant, and poorly understood complication even in optimally treated patients. As such, β-thalassemia-induced bone disease has warranted formal guidelines for management (*Thalassemia Clinical Research Network et al., 2015*).

Proposed mechanisms of bone loss in β-thalassemia include direct effects of abnormal erythroid proliferation (*Schnitzler and Mesquita, 1998*; *Gurevitch and Slavin, 2006*), increased circulating erythropoietin (Epo) (*Singbrant et al., 2011*), iron toxicity (*Weinberg, 2006*), oxidative stress (*Basu et al., 2001*), inflammation (*Lacativa and Farias, 2010*), and changes in bone marrow adiposity (*Schwartz, 2015*). Strong negative correlations between BMD and systemic iron concentrations (*Imel et al., 2016*) and the profound bone loss noted in patients with hereditary hemochromatosis (*Guggenbuhl et al., 2006*) underscore the premise that BMD and iron homeostasis may be associated causally. However, mice lacking the transferrin receptor, TFR2, display *increased* rather than decreased bone mass and mineralization despite iron overload (*Rauner et al., 2019*). These latter findings prompted us to take a fresh look at the mechanisms underpinning bone loss in diseases of iron dysregulation.

Erythroferrone (ERFE), a protein secreted by bone marrow erythroblasts, is a potent negative regulator of hepcidin (*Kautz et al., 2014*); hepcidin inhibits iron absorption and recycling (*Nemeth et al., 2005*). Thus, hepcidin suppression by increased ERFE enables an increase in iron availability during stress erythropoiesis. Very recently, ERFE has been shown to bind and sequester certain members of the bone morphogenetic protein (BMP) family, prominently BMP2, BMP6, and the BMP2/6 heterodimer (*Arezes et al., 2018*; *Wang et al., 2020*). BMPs stimulate bone formation by osteoblasts during skeletal development, modeling, and ongoing remodeling (*Hogan, 1996*). We thus hypothesized that, by modifying BMP availability, ERFE may be a key player in the newly discovered erythropoiesis-iron-bone circuitry. As a result, ERFE may also be an important link between altered iron metabolism, abnormal erythropoiesis, and bone loss in β-thalassemia.

The known mechanism of ERFE action—BMP sequestration—predicts that ERFE loss, by enhancing BMP availability, may stimulate osteoblastic bone formation (*Salazar et al., 2016*). Alternatively, recent literature shows that loss of BMP signaling increases bone mass through direct osteoclast inhibition and Wnt activation (*Broege et al., 2013*; *Jensen et al., 2009*; *Kamiya et al., 2016*; *Kamiya et al., 2008*; *Baud'huin et al., 2012*; *Gooding et al., 2019*), predicting that ERFE loss would lead to decreased bone mass. Here, we demonstrate that global deletion of *Erfe* in mice results in a low-bone-mass phenotype, which is phenocopied in β-thalassemic mice lacking ERFE (i.e. *Hbb*^th3/+; *Erfe*^-/- mice). Despite the osteopenic phenotype, we found that ERFE loss stimulated mineralization in cell culture. The net loss of bone in *Erfe*^-/- mice in the face of a pro-osteoblastic action could therefore only be attributed to a parallel increase in bone resorption, which we found was the case in

both *Erfe*$^{-/-}$ and *Hbb*$^{th3/+}$;*Erfe*$^{-/-}$ mice. Furthermore, the increase in osteoclastogenesis was osteoblast-mediated exerted via an increased expression of *Sost* and *Tnfsf11*, the gene encoding RANKL. Together, our data provide compelling evidence that ERFE loss induces BMP-mediated osteoblast differentiation, but upregulates *Sost* and *Tnfsf11* to increase osteoclastogenesis with net bone loss. Therefore, high ERFE levels in β-thalassemia are osteoprotective and prevent the bone loss when erythropoiesis is expanded.

## Materials and methods

### Mouse lines

C57BL/6 and β-thalassemic (*Hbb*$^{th3/+}$) mice (*Yang et al., 1995*) were originally purchased from Jackson Laboratories. *Erfe*$^{-/-}$ mice were a generous gift from Tomas Ganz (UCLA) (*Kautz et al., 2014*). Progeny of *Erfe*$^{-/-}$ mice crossed with *Hbb*$^{th3/+}$ yielded *Hbb*$^{th3/+}$;*Erfe*$^{-/-}$ mice. The mice have been backcrossed onto a C57BL/6 background for more than 11 generations. All mice had ad libitum access to food and water and were bred and housed in the animal facility under AAALAC guidelines. Experimental protocols were approved by the Institutional Animal Care and Use Committee at Icahn School of Medicine at Mount Sinai.

### Skeletal phenotyping

Skeletal phenotyping was conducted on 6-week- and/or 5-month-old male mice, unless otherwise noted. Mice were injected with calcein (15 mg/kg, Sigma C0875) and xylenol orange (90 mg/kg, Sigma 52097), at days 8 and 2, respectively, prior to sacrifice. Briefly, for histomorphometry, the left femur, both tibias, and L1-L3 were fixed in neutral buffered formalin (10%, v/v) for 48 hr at 4°C; transferred to sucrose (30%, w/v) at 4°C overnight; and embedded and sectioned at –25°C (5–6 µm thick sections, 10X) (*Dyment et al., 2016*). Unstained sections were analyzed by fluorescence microscopy (Leica Upright DM5500) to determine the mineralizing surface and interlabel distance using image J. Von Kossa staining of sections was used to quantify fractional bone volume (BV/TV) and trabecular thickness (Tb.Th). Tartrate-resistant acid phosphatase (ACP5) staining (Sigma 387A) was used to identify osteoclasts, counterstained with aniline blue using Olympus Stereoscope MVX10 (1X). Images were analyzed by TrapHisto and OsteoidHisto (*van 't Hof et al., 2017*). On the day of sacrifice, BMD was also measured in intact mice (*Shi et al., 2016*). Frozen bone sections were incubated for 4 min at room temperature in Alkaline Phosphatase Substrate Solution ImmPACT Vector Red (Vector Laboratories). After washing with buffer, the sections were counterstained with hematoxylin (Vector Laboratories) and mounted with VectaMount AQ Mounting Medium (Vector Laboratories). Sections were visualized using Olympus BH-2 Microscope and images obtained with OMAX A35180U3 Camera were analyzed by ImageJ.

### Isolation and culture of bone marrow cells

For osteoblast cultures, fresh bone marrow cells were seeded in 12-well plates (0.6 × 10$^6$ cells *per* well) under differentiating conditions (αMEM, 10% FBS, 1% penicillin/streptomycin, 1 M β-glycerol phosphate, and 0.5 M ascorbic acid) for 21 days to induce the formation of mature, mineralizing osteoblast colonies, Cfu-ob, as before (*Maridas et al., 2018*). Cultured osteoblasts were treated with BMP2 (50 ng/ml) or BMP6 (50 ng/ml) for 30 min on day 3 of culture to assess effects on BMP-mediated signaling, as previously (*Rauner et al., 2019*). For osteoclast cultures, bone marrow hematopoietic stem cells (non–adherent) from wild type and *Erfe*$^{-/-}$ were seeded in six-well plates (10$^6$ cells *per* well) in the presence of αMEM, 10% FBS, 1% penicillin/streptomycin and M-CSF (25 ng/mL, PeproTech) for 48 hr, followed by the addition of RANK-L (50 ng/mL, PeproTech) for 5 days. In experiments testing Epo responsiveness, 20 U/ml of Epo was added for the duration of the differentiation process, for 21 and 5 days in osteoblast and osteoclast cultures, respectively.

Erythroblasts were isolated from bone marrow and purified using CD45 beads, as previously (*Li et al., 2017*) with minor modifications. Briefly, mouse femur was flushed, single-cell suspensions incubated with anti-CD45 magnetic beads (Mylteni), and erythroid lineage-enriched cells that flowed through the column were collected. Erythroid-enriched cells were incubated with anti-mouse TER119-phycoerythrin Cy7 (PE-Cy7) (BioLegend) and CD44-allophycocyanin (APC) (Tonbo, Biosciences). Non-erythroid and necrotic cells were identified and excluded from analyses using anti-CD45

(BD Pharmigen), anti-CD11b, and anti-Gr1 (APC-Cy7) (Tonbo, Biosciences) antibodies. Erythroid precursors were selected by gating and analyzed using TER119, CD44, and forward scatter as previously described (*Li et al., 2017*). Samples were analyzed on either FACSCanto I or LSRFortessa flow cytometer (BD Biosciences). To determine levels of *Erfe* mRNA expression in Epo–stimulated conditions, erythroblasts were cultured in the presence or absence of 20 U/ml Epo for 15 hr as described (*Kautz et al., 2014*).

## Primary hepatocyte culture

Hepatocytes were isolated by perfusion with collagenase and liver digestion, as described previously (*Merlin et al., 2017*). Briefly, 0.025% (w/v) collagenase type IV (Gibco) and 5 mM $CaCl_2$ was added to Leffert perfusion buffer containing 10 mM HEPES, 3 mM KCl, 130 mM NaCl, 1 mM $NaH_2PO_4$. $H_2O$, and 10 mM D-glucose (Sigma). Live cells were purified by Percoll (Sigma) and plated in six-well plates ($0.25 \times 10^6$ cells *per* well) in William's Medium E (Sigma) supplemented with antibiotics and 5% fetal bovine serum (FBS) for 2 hr to allow the hepatocytes to attach. Cells were starved overnight with William's Medium E lacking FBS, and were then treated for 6 hr with conditioned or control media from wild type or *Erfe*$^{-/-}$ osteoblast and osteoclast cultures (day 6 and day 5, respectively) in the presence of 50% (v/v) William's Medium E and 5% FBS.

## Quantitative PCR

RNA was purified from osteoblasts, osteoclasts, erythroblasts, and hepatocytes using PureLink RNA (Sigma) and analyzed with SuperScript III Platinum SYBR Green One-Step (Invitrogen). As previously described (*Koide et al., 2017*; *Dumas et al., 2008*), ΔΔCT values were used to calculate fold increases relative to β-actin, α-tubulin, and RLP4. Primers are listed in *Table 1*.

## Western immunoblotting and ELISA

For western immunoblotting, differentiated cells at day three were lysed in ice cold SDS page lysis buffer (2% SDS, 50 mM Tris-HCl, pH 7.4, 10 mM EDTA) with protease and phosphatase inhibitors. 20 μg of heat–denatured protein was loaded onto a 10% gel, run, and transferred onto a 0.4 μm nitrocellulose membrane (Thermo Scientific). After blocking with 5% BSA in Tris–buffered saline with 1% Tween-20 (TBS-T), the membranes were incubated with primary antibodies to signaling proteins (*Table 2*) overnight at 4°C, washed, and incubated with the corresponding HRP–conjugated secondary antibodies at room temperature. Proteins were visualized using the ImageQuant LAS 4010 and quantified using Image J. Osteoblast supernatants from wild type and *Erfe*$^{-/-}$ mice were collected and centrifuged for 10 min at 10,000 x g, and BMP2 (Abnova) and RANKL (R and D) concentrations were measured by ELISAs. Serum BMP2 concentration was determined using mouse BMP2 ELISA

**Table 1.** Primers used in the presented studies.

| Gene | Forward (sense) | Reverse (antisense) |
|------|-----------------|---------------------|
| *Actb* | TTCTTTGCAGCTCCTTCGTT | ATGGAGGGGAATACAGCCC |
| *Tubb* | CTGGAGCAGTTTGACGACAC | TGCCTTTGTGCACTGGTATG |
| *Hamp* | TGAGCAGCACCACCTATCTC | ACTGGGAATTGTTACAGCATTT |
| *Acp5* | ACCTGTGCTTCCTCCAGGAT | TCTCAGGGTGGGAGTGGG |
| *Ctsk* | CCATATGTGGGCCAGGATG | TCAGGGCTTTCTCGTTCCC |
| *Runx2* | GTGGCCACTTACCACAGAGC | GTTCTGAGGCGGGACACC |
| *Alpl* | ACACCTTGACTGTGGTTACTGCTGA | CCTTGTAGCCAGGCCCGTTA |
| *Osx* | TGAGGAAGAAGCCCATTCAC | GTGGTCGCTTCTGGTAAAGC |
| *Col1a1* | CCTGGCAAAGACGGACTCAAC | GCTGAAGTCATAACCGCCACTG |
| *Tnsf11* | CAGCCATTTGCACACCTCAC | GTCTGTAGGTACGCTTCCCG |
| *Opg* | ACAGTTTGCCTGGGACCAAA | CAGGCTCTCCATCAAGGCAA |
| *Dmp1* | GGGCTGTGTTGTGCAAGACA | GGTGCACACCTGACCTTCTTTAA |
| *Fam132b* | ATGGGGCTGGAGAACAGC | TGGCATTGTCCAAGAAGACA |

**Table 2.** Antibodies used in the presented studies.

| Antibody | Company | # catalog | Dilution | BSA/Milk (5%) | Rabbit/mouse |
|---|---|---|---|---|---|
| *PRIMARY antibodies* | | | | | |
| pSMAD 1/5/8 | Cell signaling | 9511 | 1:1000 | BSA | Rabbit |
| pSMAD 1/5/8 (monoclonal) | Cell signaling | 9516 | 1:1000 | BSA | Rabbit |
| SMAD 1 | Cell signaling | 6944S | 1:1000 | BSA | Rabbit |
| p-ERK (monoclonal) | Cell signaling | 4376 | 1:1000 | BSA | Rabbit |
| ERK | Cell signaling | 4695 | 1:1000 | BSA | Rabbit |
| pp38 | Cell signaling | 4511 | 1:1000 | BSA | Rabbit |
| p38 | Cell signaling | 8690 | 1:1000 | BSA | Rabbit |
| Beta-actin | ThermoFisher | MA515452 | 1:1000 | BSA | Mouse |
| *SECONDARY antibodies* | | | | | |
| Rabbit | Cell signaling | 7074 | 1:5000 | BSA | |
| Mouse | GE Healthcare | NXA931V | 1:20000 | BSA | |

(abnova, KA0542), *per* manufacturers instructions. ERFE concentration in conditioned media was determined as described (*Kautz et al., 2015*) with the substitution of DELFIA europium–conjugated streptavidin for horseradish-peroxidase-conjugated streptavidin. Fluorescence was measured by CLARIOstar plate reader.

## Complete blood counts

Peripheral blood (100 µL from each mouse) was collected from the retro-orbital vein in EDTA-coated tubes and analyzed by IDEXX Procyte Hematology Analyzer.

## Statistical analyses

Data are reported as means ± SEM. Unpaired Student's t-test was used to determine if differences between groups were significant at $p < 0.05$.

## Results

To understand if ERFE has a role in regulating skeletal integrity in health, we first studied the effect of ERFE loss on BMD and bone remodeling in adult *Erfe*[-/-] mice, as well as in compound mutant mice in which the *Erfe* gene was deleted on a β-thalassemia *Hbb*[th3/+] background. Compared with wild-type littermates, both 6-week-old and 5-month-old male *Erfe*[-/-] mice showed significant reductions in whole body BMD, and BMD at mainly cortical (femur and tibia) sites (*Figure 1A and B*). However, in contrast to young mice, the older *Erfe*[-/-] mice did not show a difference in lumbar spine BMD compared with wild-type littermates. Interestingly, unlike hypogonadal bone loss, which is predominantly trabecular, the sustained reduction in femur and tibia BMD is consistent with prominent cortical loss seen in patients with β-thalassemia (*Haidar et al., 2011*; *Vogiatzi et al., 2006*).

Bone resorption and bone formation are tightly coupled to maintain bone mass during each remodeling cycle (*Zaidi, 2007*). Bone is lost when either both processes are increased—with resorption exceeding formation, as in hypogonadism—or when there is uncoupling in which formation decreases while resorption rises, as in glucocorticoid excess (*Zaidi, 2007*). To differentiate between relative increases and uncoupling, we measured both formation and resorption in intact bone. Dynamic histomorphometry performed after the sequential injections of calcein and xylenol orange, which yielded dual fluorescent labels, allowed us to derive parameters of bone formation. We observed that mineralizing surface (MS), mineral apposition rate (MAR) and bone formation rates (BFR) were all increased in young *Erfe*[-/-] mice, consistent with the pro-osteoblastic (anabolic) action of ERFE deficiency (see below) (*Figure 1C and D*). No differences in MS, MAR, and BFR were noted in 5-month-old mice (*Figure 1D*). We also analyzed alkaline phosphatase stained sections of femurs

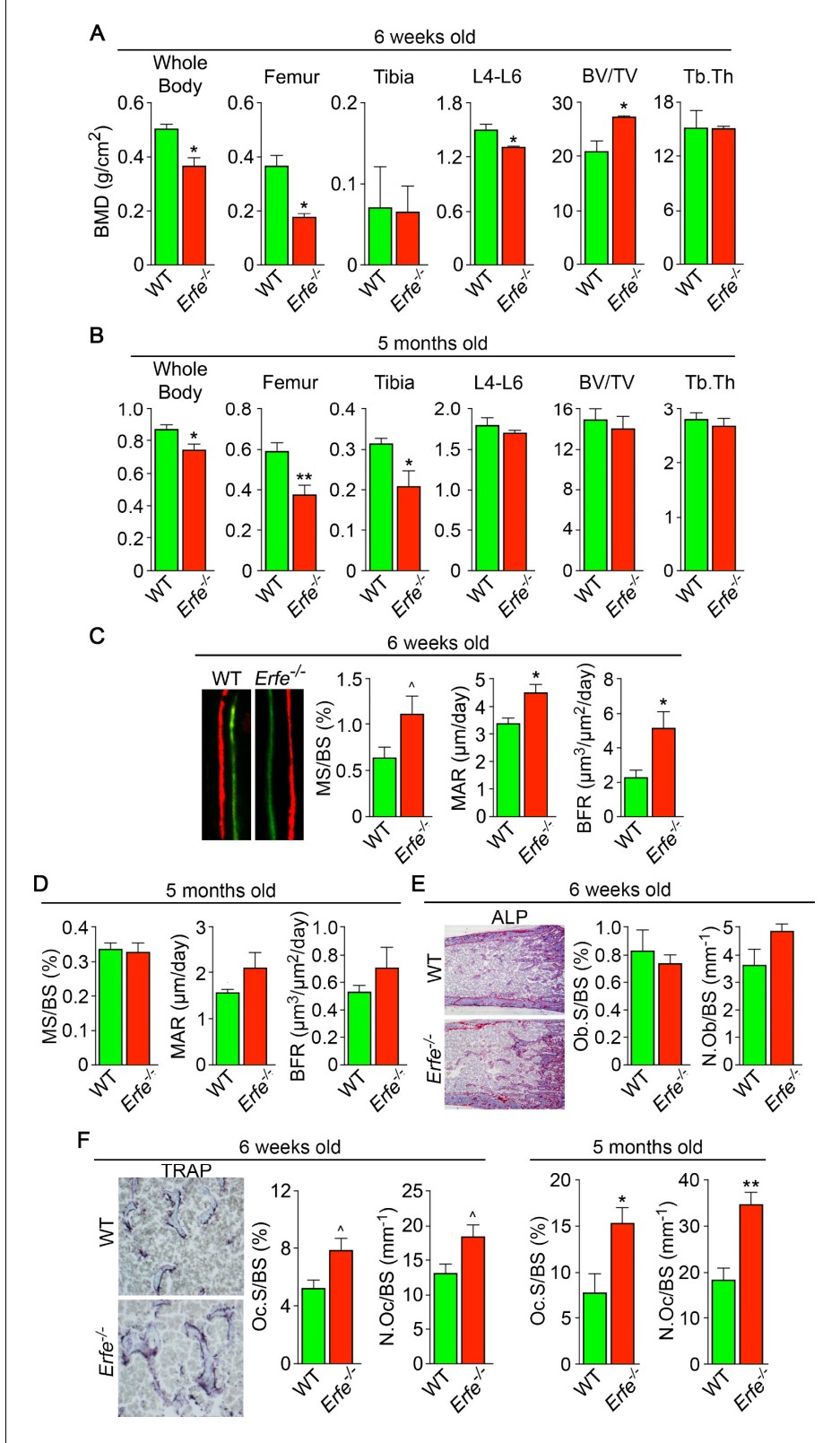

**Figure 1.** ERFE loss results in high turnover osteoporosis. Bone mineral density (BMD) measured in whole body, femur, tibia, and lumbar spine (L4–L6) along with bone volume (BV/TV) and trabecular thickness (Tb.Th) in growing (6-week-old) (A) and mature (5-month-old) (B) *Erfe*⁻/⁻ and wild-type (WT) littermates. Dynamic histomorphometry following two i.p. injections of calcein (green) and xylenol orange (red) given at days 8 and 2, respectively. *Figure 1 continued on next page*

*Figure 1 continued*

Representative dual labels from the epiphysis are shown, together with measured and derived parameters, namely mineralizing surface (MS) as a function of bone surface (BS), mineral apposition rate (MAR) and bone formation rate (BFR) in 6-week-old (**C**) and 5-month-old (**D**) mice. (**E**) Alkaline phosphatase staining (magenta) in sections of femura demonstrates no differences in osteoblast surface (Ob.S) and number (N.Ob) as a function of BS in 6-week-old *Erfe*⁻/⁻ and WT mice. (**F**) TRAP staining at the epiphysis showing both osteoclast surface (Oc.S) and number (N. Oc) as a function of BS. Statistics: Mean ± SEM; unpaired two-tailed Student's t-test; *p<0.05, **p<0.01, ʹ0.05 < p < 0.1, *N* = 3–6 mice per group.

The online version of this article includes the following figure supplement(s) for figure 1:

**Figure supplement 1.** Erythropoiesis-related parameters in Erfe-/- mice.

---

to find no difference in osteoblast surfaces (Ob.S) or osteoblast number (N.Ob) *per* bone surface (BS) in 5–week–old *Erfe*⁻/⁻ relative to wild type littermates (*Figure 1E*).

Finally, to study whether an increase in osteoclastic bone resorption caused the notable reduction in BMD in *Erfe*⁻/⁻ mice, we measured TRAP-positive osteoclast surfaces (Oc.S) and number (N.Oc) *per* bone surface (BS). Both Oc.S/BS and N.Oc/BS were increased significantly in *Erfe*⁻/⁻ compared with wild type bones in older mice, and to a lesser extent, in younger mice (*Figure 1F*). Thus, the overall low-bone-mass phenotype in *Erfe*⁻/⁻ mice primarily resulted from a relative increase in osteoclastic bone resorption over osteoblastic bone formation, suggesting that ERFE has a function in preventing skeletal loss. To confirm that decreased BMD in *Erfe*⁻/⁻ mice did not result from changes in erythropoiesis, we measured circulating red blood cells (RBCs) and reticulocytes, and bone marrow erythroblasts. We also measured spleen weight given the ubiquity of compensatory erythropoiesis that results in splenomegaly. Our results show no differences between 6-week-old wild type and *Erfe*⁻/⁻ mice (*Figure 1—figure supplement 1*), consistent with what has been previously reported in *Erfe*⁻/⁻ mice (*Kautz et al., 2014*).

To probe the mechanism of action of ERFE on osteoblastic bone formation and osteoclastic bone resorption, we first asked which cells in bone marrow produce ERFE, and whether secreted ERFE was functional. Intriguingly, time course studies in differentiating osteoblasts revealed that *Erfe* expression was 10- and twofold higher at 3 and 21 days of culture, respectively, compared with cultured erythroblasts—the only previously known source of ERFE in bone marrow (*Figure 2A*). To confirm that cultured cells were of the osteoblastic lineage, we evaluated *Alp* expression, a known marker of osteoblast lineage, and demonstrate increased *Alp* expression as early as day 3 in culture (*Figure 2—figure supplement 1*). Furthermore, *Erfe* expression in mature osteoclasts was similar to cultured erythroblasts, with little expression in immature osteoclasts (*Figure 2B*). Likewise, conditioned media from osteoblast cultures revealed increased ERFE concentration at 3 days with no differences in conditioned media from osteoclast cultures (*Figure 2C*).

To determine whether osteoblast-derived ERFE was functional, we established a bioassay based on the known inhibitory action of ERFE on hepcidin (*Hamp*) expression. For this, wild-type hepatocytes were exposed to supernatants from differentiating wild type or *Erfe*⁻/⁻ osteoblasts. *Hamp* expression was suppressed with *Erfe*⁻/⁻ osteoblast supernatants, but importantly, this suppression was significantly greater with wild type supernatants (*Figure 2D*). No *Hamp* suppression was evident with wild type or *Erfe*⁻/⁻ osteoclast supernatants. This latter suggests that osteoblast– but not osteoclast-derived ERFE is functional. However, as *Erfe*⁻/⁻ supernatants also suppressed *Hamp* expression, other yet unknown osteoblast-derived factors likely function in hepcidin regulation. Finally, unlike in erythroblasts, *Erfe* expression in mature osteoblasts or osteoclasts was not responsive to Epo (*Figure 2E*).

Given that osteoblasts secrete ERFE that is known to inhibit hepcidin (*Kautz et al., 2014*) by sequestering BMPs (*Arezes et al., 2018*; *Wang et al., 2020*; *Arezes et al., 2020*) that are skeletal anabolics (*Hogan, 1996*), we measured serum BMP2 concentration to find elevated BMP2 levels in *Erfe*⁻/⁻ relative to wild-type mice (*Figure 3A*). Given the specific importance of BMP2 in bone remodeling (*Salazar et al., 2016*), these results are consistent with the previously demonstrated sequestration of BMP2, along with BMP6, by ERFE (*Wang et al., 2020*)—namely, loss of ERFE led to decreased BMP sequestration. We thus hypothesized that ERFE functions in modulating bone formation by sequestering BMPs and tested whether the loss of ERFE facilitates BMP2-mediated signaling in the osteoblast in vitro. We found that the concentration of BMP2 was higher in supernatants from

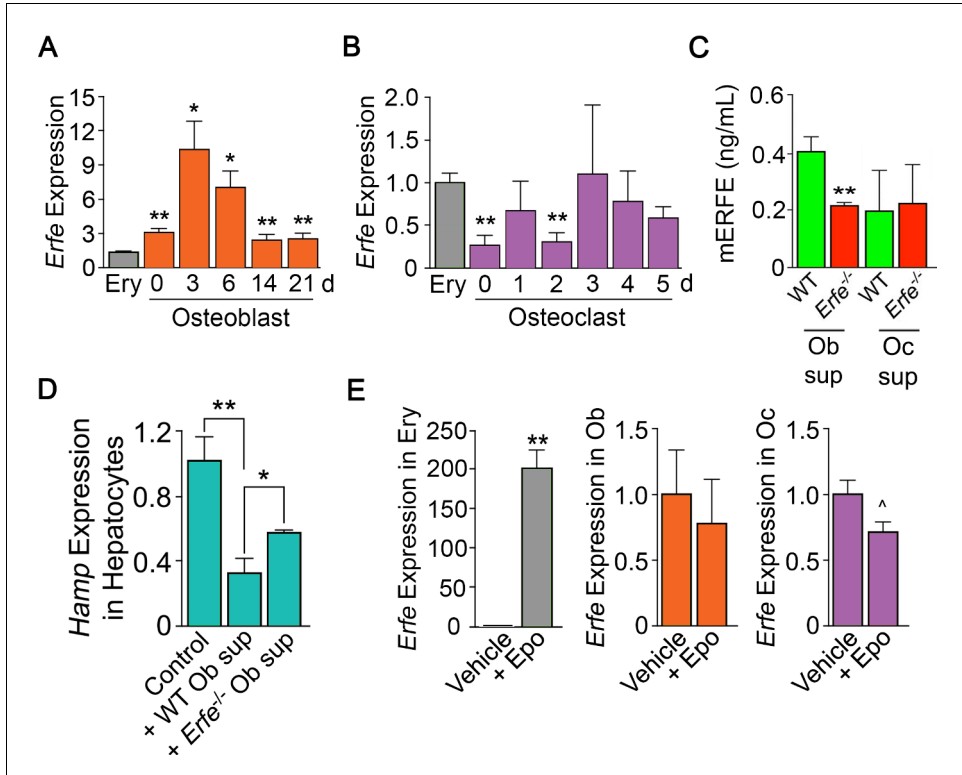

**Figure 2.** ERFE is expressed at higher levels in osteoblasts than in erythroblasts. (A) Quantitative PCR showing high levels of *Erfe* expression in osteoblasts from wild type mice cultured under differentiating conditions. Notably, at 3 days of culture, there was a 10-fold greater expression in osteoblasts relative to bone-marrow-derived wild-type cultured erythroblasts. (B) Quantitative PCR showing comparable levels of *Erfe* expression in osteoclasts at 3–5 days of culture relative to bone-marrow-derived wild-type cultured erythroblasts from wild-type mice cultured under differentiating conditions. (C) Increased supernatant murine ERFE (mERFE) concentration in day 3 osteoblast cultures and no difference in day 5 osteoclast cultures from wild type relative to *Erfe*$^{-/-}$ mice (detection limit of 0.2 ng/ml mERFE). (D) Hepcidin (HAMP) expression is suppressed in primary wild-type hepatocytes in response to conditioned media from wild-type relative to *Erfe*$^{-/-}$ osteoblast cultures (day 6), confirming functionality of osteoblast-derived ERFE. Control hepatocytes were exposed to osteoblast culture media. (E) Unlike in erythroblasts, *Erfe* expression in cultured wild-type osteoblasts and osteoclasts does not respond to erythropoietin (Epo). Statistics: Mean ± SEM; unpaired two-tailed Student's t-test; *p<0.05, **p<0.01, 0.05 < *P* < 0.1, N = 3 wells per group.

The online version of this article includes the following figure supplement(s) for figure 2:

**Figure supplement 1.** Alkaline phosphatase expression increased during osteoblast differentiation in culture.

cultured *Erfe*$^{-/-}$ osteoblasts compared with wild type cultures (*Figure 3B*). Consistent with this difference, BMP2-activated signaling pathways, namely phosphorylated Smad1/5/8 and ERK1/2, but not phosphorylated p38, were enhanced in *Erfe*$^{-/-}$ compared with wild-type osteoblasts (*Figure 3C*).

To further understand how ERFE impacts BMP2-mediated signaling, we evaluated the effect of BMP2 on wild type and *Erfe*$^{-/-}$ osteoblasts in vitro. Treatment with BMP2 (50 ng/ml) in osteoblast cultures showed that pSmad1/5/8 and pERK signaling was not further induced in *Erfe*$^{-/-}$ relative to wild-type osteoblasts (*Figure 3D and E*). In all, the data establish that increased BMP2 in *Erfe*$^{-/-}$ mice leads to maximal induction of BMP signaling that remains unaffected by the further addition of BMP2. To test whether an ERFE effect on bone is BMP2 specific, we also repeated these experiments using BMP6, demonstrating results similar to the effects of BMP2 on BMP signaling pathways in wild type and *Erfe*$^{-/-}$ osteoblasts in vitro (*Figure 3—figure supplement 1*). These findings support the hypothesis that ERFE functions in bone by sequestering BMPs, thus, attenuating downstream signaling.

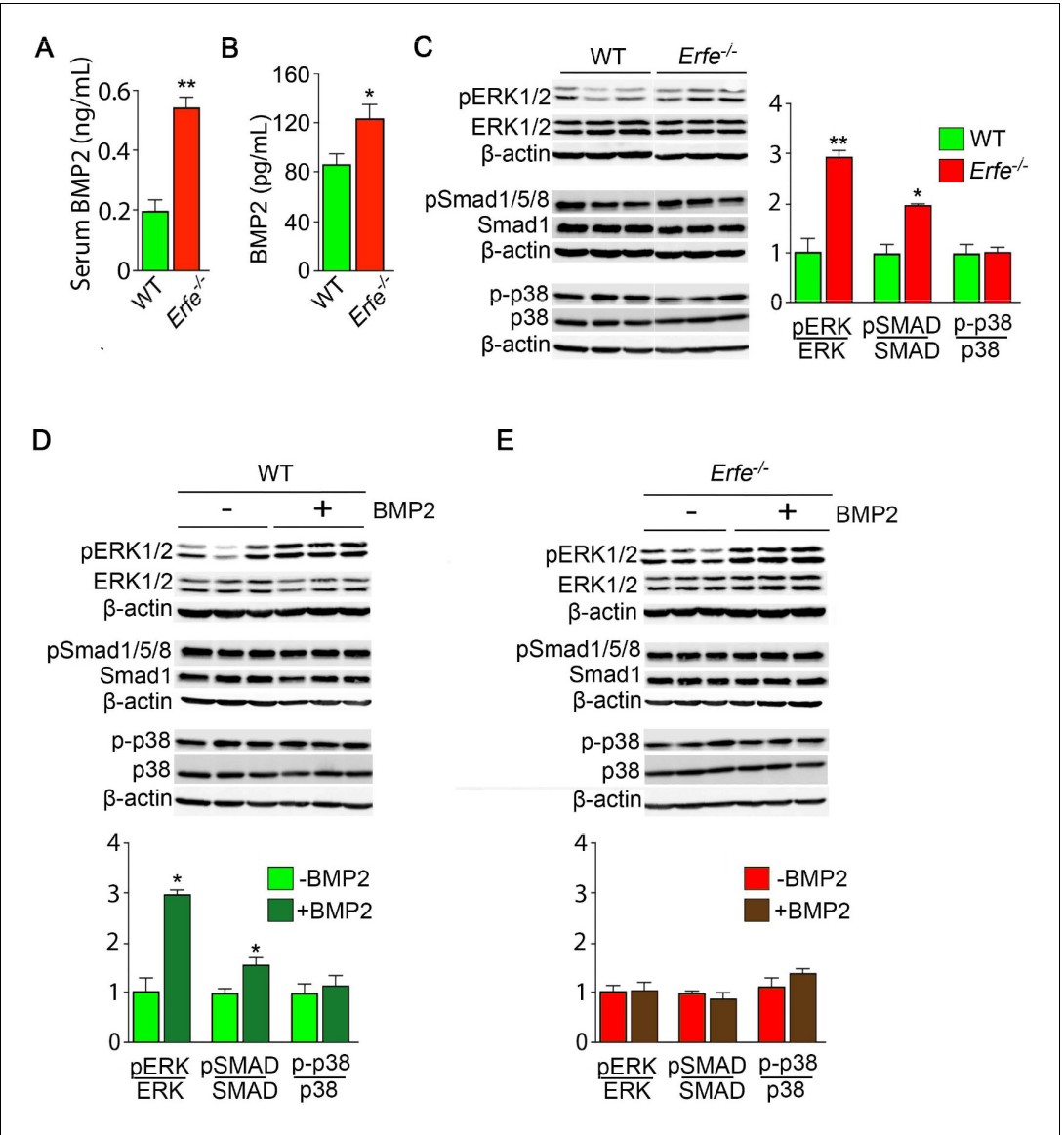

**Figure 3.** ERFE function on bone involves BMP-2 sequestration. (**A**) BMP2 ELISA demonstrates elevated BMP2 concentration in serum samples from *Erfe*<sup>-/-</sup> relative to WT mice (*N* = 4 per group). In the 3-day cultures, there was an increase in BMP2 concentration in culture supernatants from *Erfe*<sup>-/-</sup> relative to WT osteoblasts (*N* = 6 per group) (**B**). (**C**) Similarly, signaling via the known BMP receptor pathways, namely ERK1/2 and Smad1/5/8, without changes in p38/MAPK, increase in *Erfe*<sup>-/-</sup> relative to WT osteoblasts; western blots with quantification shown. Finally, pSmad1/5/8 and pERK1/2 signaling is further induced by BMP2 (50 ng/ml) only in WT (**D**) but not in *Erfe*<sup>-/-</sup> (**E**) osteoblasts. Statistics: Mean ± SEM; unpaired two-tailed Student's t-test; *p<0.05, **p<0.01.

The online version of this article includes the following figure supplement(s) for figure 3:

**Figure supplement 1.** ERFE function on bone involves BMP-6 sequestration.

We studied whether the stimulation of bone formation in *Erfe*<sup>-/-</sup> mice was due to a cell–autonomous action of ERFE on osteoblasts. For this, we compared the ability of wild type and *Erfe*<sup>-/-</sup> bone marrow stromal cells ex vivo to differentiate into mature mineralizing colony forming units–osteoblastoid (Cfu-ob). Stromal cells from 5-month-old *Erfe*<sup>-/-</sup> mice showed enhanced von Kossa staining of mineralizing Cfu-ob colonies (*Figure 4A*). This mineralizing phenotype was associated with enhanced expression of the osteoblast transcription factors *Runx2* and *Sp7*, and downstream genes *Sost* and *Tnfsf11* (*Yang et al., 2010*; *Pérez-Campo et al., 2016*), increased supernatant RANKL levels, and suppressed expression of *Opg* (*Figure 4B and C*). Enhanced RANKL profoundly increases

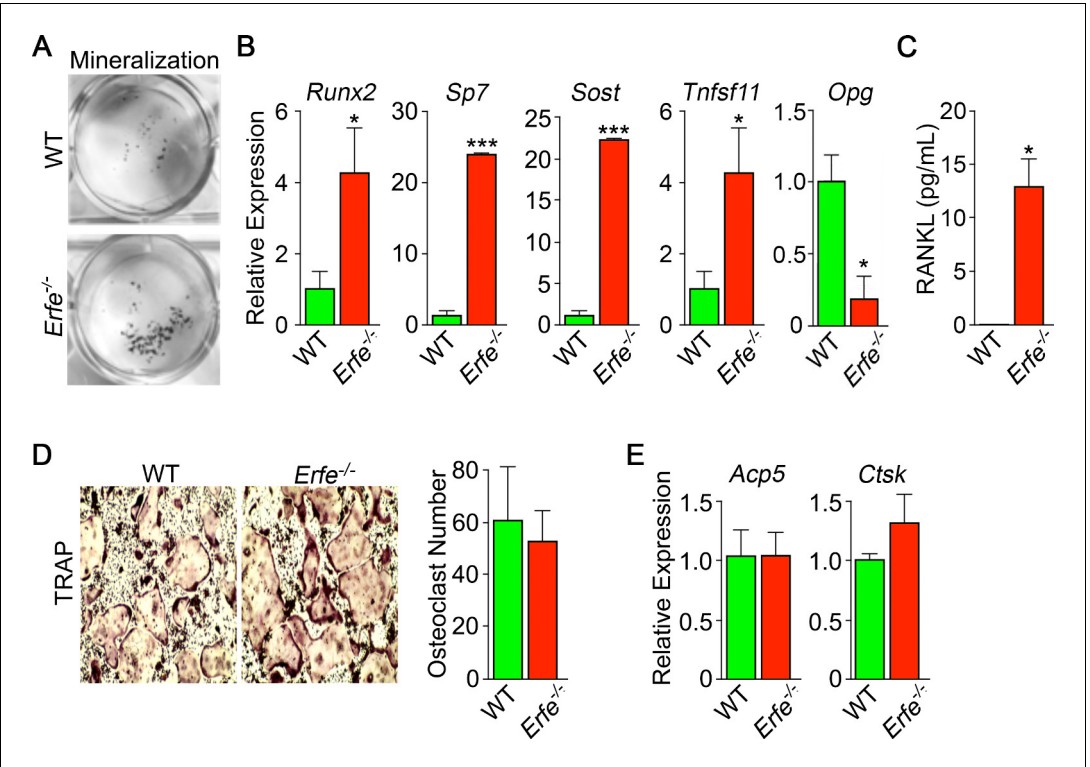

**Figure 4.** Mechanism of action of ERFE on bone involves interplay between osteoblastic RANKL and sclerostin. Osteoblasts from 5-month-old wild type and *Erfe*[-/-] mouse bone marrow cultured under differentiating conditions for 5 or 21 days. Loss of ERFE resulted in accelerated mineralization, noted by an increase in Von Kossa-stained nodules (A). Consistent with the cellular phenotype is the upregulation in *Erfe*[-/-] osteoblasts of *Runx2*, *Sp7*, *Sost*, and *Tnfsf11* expression and suppression of *Opg* expression (quantitative PCR on 21-day cultures) (B) as well as increased secreted RANKL (ELISA on 3 day cultures) (C). In vitro osteoclastogenesis assays show that ERFE loss does not alter osteoclast number, as measured by TRAP staining (D), or the expression of osteoclast genes, namely *Acp5* or *Ctsk* (E). Statistics: Mean ± SEM; unpaired two-tailed Student's t-test; *p<0.05, **p<0.01; wells per group – three for A-C.

osteoclastogenesis, as noted below, while sclerostin, encoded by the *Sost* gene, reduces the production of OPG, hence further increasing osteoclast formation.

Given that *Erfe* is expressed in osteoclasts, and that *Erfe*[-/-] mice display a pro–resorptive phenotype, we questioned whether ERFE directly affected the osteoclast, or whether the action resulted via a primary osteoblastic effect. *Erfe*[-/-] bone marrow cell cultures derived from 5-month-old mice showed no difference in TRAP-positive osteoclast number compared to wild-type cultures (*Figure 4D*). Consistent with this, the program of osteoclast gene expression remained unchanged in these 5-day cultures (*Figure 4E*). The data collectively suggest that the absence of ERFE results in the de-sequestration of BMP2, stimulates the osteoblast to upregulate RANKL and sclerostin, and thus enhances osteoclastic bone resorption indirectly.

Finally, we explored whether ERFE mediates osteoprotection in *Hbb*[th3/+] mice, a model of human NTDT given that *Erfe* is upregulated in *Hbb*[th3/+] marrow erythroblasts (*Vogiatzi et al., 2006*; *Kautz et al., 2014*; *Li et al., 2017*; *Kautz et al., 2015*; *Vogiatzi et al., 2010*). We crossed *Hbb*[th3/+] mice with *Erfe*[-/-] mice to generate *Hbb*[th3/+];*Erfe*[-/-] compound mutants. Whole body and site-specific measurements at mainly cortical sites, namely femur and tibia, and vertebral trabecular (L4-L6) bone showed striking reductions in BMD in 5-month-old *Hbb*[th3/+];*Erfe*[-/-] mutants compared with *Hbb*[th3/wt] mice, most notably in cortical bone (*Figure 5A*). The trabecular bone loss was consistent with reduced histomorphometrically determined fractional bone volume (BV/TV) and trabecular thickness (Tb.Th) in the femoral epiphyses (*Figure 5B*). There was a trend toward increases in MAR (*Figure 5C*), but a significant increase in TRAP-positive N.Oc and Oc.S in *Hbb*[th3/+];*Erfe*[-/-] bones

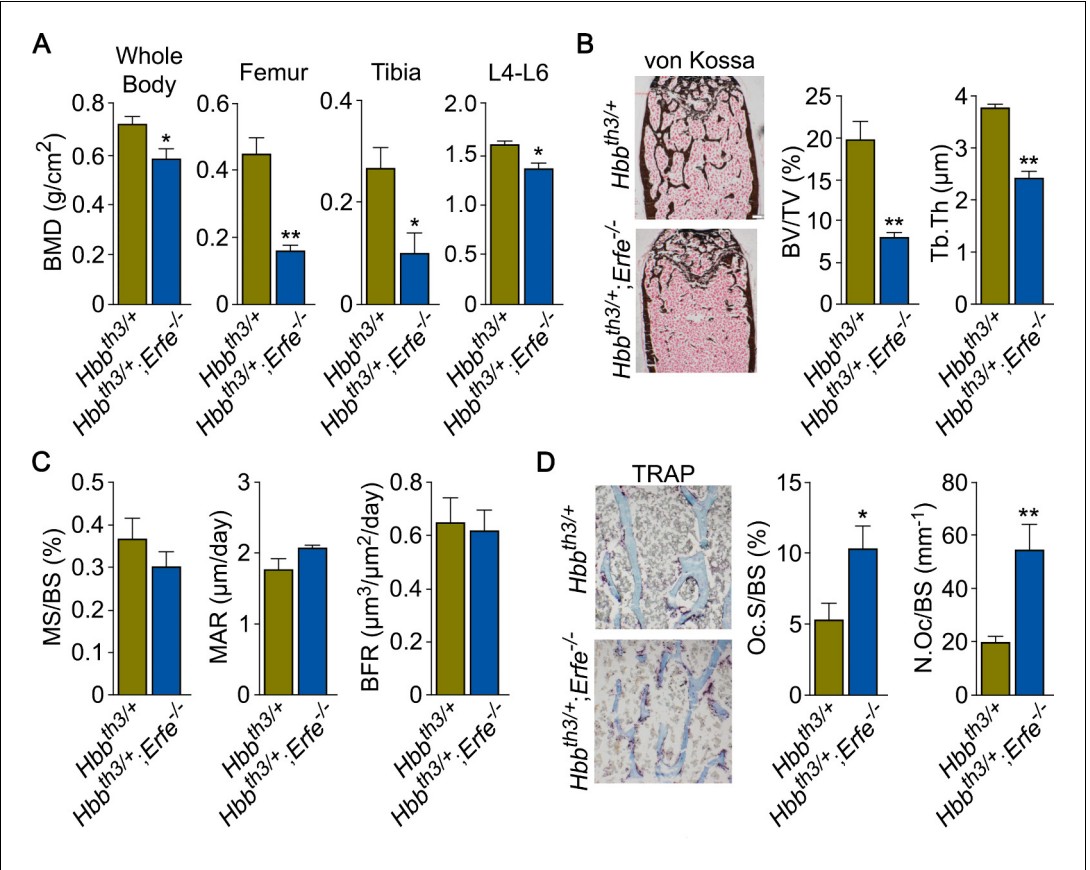

**Figure 5.** ERFE loss in β-thalassemia mice causes profound bone loss. (**A**) Bone mineral density (BMD) measured in whole body, femur, tibia, and lumbar spine (**L4–L6**) in 5-month-old β-thalassemia mice (*Hbb*^th3/+ mice) and compound *Hbb*^th3/+;*Erfe*^-/- mutants. (**B**) Representative section of femoral epiphyses stained with Von Kossa, and quantitative estimates of bone volume (BV/TV) and trabecular thickness (Tb.Th). (**C**) Dynamic histomorphometry following two i.p. injections of calcein (green) and xylenol orange (red) given at days 8 and 2, respectively. Shown are measured and derived parameters, namely mineralizing surface (MS), mineral apposition rate (MAR) and bone formation rate (BFR). (**D**) Representative image of TRAP (ACP5) staining of femoral epiphysis, also showing both osteoclast surface (Oc.S) and number (N.Oc), expressed as a function of bone surface (BS). Statistics: Mean ± SEM; unpaired two-tailed Student's t-test; *p<0.05, **p<0.01; N = 4–5 mice per group.

The online version of this article includes the following figure supplement(s) for figure 5:

**Figure supplement 1.** Erythropoiesis-related parameters in Hbb^th3/+ and Hbb^th3/+;*Erfe*^-/-mutant mice.

compared with wild-type controls (*Figure 5D*)—changes expected to produce reduction in bone mass. These findings document ERFE-mediated skeletal protection in β-thalassemia.

To confirm that decreased BMD in *Hbb*^th3/+;*Erfe*^-/- mice did not result from further expanded erythropoiesis, we measured circulating RBCs and reticulocytes, bone marrow erythroblasts, and spleen weight. Our results demonstrate a mildly decreased RBC count and hemoglobin, but no differences in spleen weight or bone marrow erythroblasts between 6-week-old *Hbb*^th3/+ and *Hbb*^th3/+; *Erfe*^-/- mice (*Figure 5—figure supplement 1*). This is consistent with what has been previously reported in *Hbb*^th3/+;*Erfe*^-/- mice (*Kautz et al., 2015*).

## Discussion

To date, the only known function of ERFE was on hepatocellular hepcidin expression exerted through the sequestration of BMPs (*Arezes et al., 2018*; *Wang et al., 2020*). Using genetically–modified mice and in vitro assays, we identify a new role for ERFE in skeletal protection. First, we show that *Erfe* expression is higher in osteoblasts comapred with erthyroblasts. Second, we find that

ERFE is a potent down-regulator of BMP2–mediated signaling and RANK-L production by osteoblasts. Third, ERFE loss in vivo enhances bone formation, while also stimulating resorption by inducing the expression of osteoblastic *Tnfsf11* and *Sost.* The net effect of these opposing changes is bone loss in both young and old mice (*Figure 6*). Fourth, although also produced by osteoclasts, ERFE displays no cell-autonomous actions on osteoclast function. Taken together, and consistent with prior inferences (*Broege et al., 2013*; *Jensen et al., 2009*; *Kamiya et al., 2016*; *Kamiya et al., 2008*; *Baud'huin et al., 2012*; *Gooding et al., 2019*), ERFE functions to protect the skeleton by negatively regulating BMP signaling in osteoblasts, with indirect inhibitory effects on osteoclastic bone resorption.

Bone is a highly dynamic and purposefully organized tissue which undergoes constant remodeling in response to changing metabolic and mechanical needs. Bone remodeling is a process in which bone resorption by osteoclasts is balanced by synthesis of new bone by osteoblasts, which then undergo terminal differentiation to become mechanosensory osteocytes. Multiple local cytokines and systemic hormones regulate the delicate balance of bone resorption and bone formation, enabling bone cells to communicate among themselves as well as with other cells in the bone marrow. For example, osteocytes secrete Sclerostin, encoded by the SOST gene, which, in turn, inhibits further osteoblast differentiation. Osteoblasts and osteocytes also secrete RANKL and OPG. Osteoclasts express RANK, the RANKL receptor, and binding stimulates the differentiation of osteoclast precursors into mature osteoclasts; OPG sequesters RANKL to prevent unrestricted osteoclast differentiation. SOST optimizes the relative proportion of RANKL and OPG to induce bone resorption. Our findings demonstrate that while ERFE loss leads to increased bone mineralization in vitro and bone formation increases as expected with age, the composite effect in vivo results in greater enhancement of osteoclastogenesis relative to osteoblastogenesis in $Erfe^{-/-}$ relative to wild-type mice, consequently decreasing BMD (*Figure 6*).

We intentionally compared growing (6-week-old) and mature (5-month-old) mice to assess the potentially distinct or cumulative effect of ERFE loss on bone growth and/or remodeling, respectively. Our results demonstrate that ERFE loss leads to impaired BMD in both cohorts of mice. However, while MS/BS, MAR, BFR, and BV/TV are increased in 6-week-old $Erfe^{-/-}$ relative to wild-type mice, no differences are evident between 5-month-old $Erfe^{-/-}$ and wild-type mice. These findings strongly suggest that over time, while both bone formation and resorption increase relative to younger mice, the balance between them favors bone resorption (*Figure 6C*). This interpretation is further corroborated by a more profound increase in osteoclast surface and osteoclast number in 5-month-old $Erfe^{-/-}$ relative to wild type than in 6-week-old mice.

We show that supernatants from wild-type osteoblast cultures suppress *Hamp* expression, suggesting that BMP sequestration is likely a common mechanism that underpins both the hepatocellular and skeletal actions of ERFE. Recombinant ERFE binds certain bone-active BMPs, namely BMP2, BMP6, and the BMP2/6 heterodimer (*Arezes et al., 2018*; *Wang et al., 2020*), of which BMP2 is most relevant to adult bone formation (*Salazar et al., 2016*). We posit that ERFE is a negative regulator of osteoblastic bone formation, and that the absence of ERFE increases bioavailable BMP to promote osteoblast differentiation. Our results indeed demonstrate that BMP2 levels are elevated in supernatants from cultured $Erfe^{-/-}$ osteoblasts, with enhanced downstream signals, notably phosphorylated Smad1/5/8 and ERK1/2. In all, the findings provide strong support for BMP sequestration as the mechanism of action of ERFE on bone.

We have also used the β-thalassemia mouse, $Hbb^{th3/+}$, as a relevant disease model to study a role for ERFE in β-thalassemia, a condition with known elevations in ERFE. The results presented reflect evidence derived from analysis of male mice. We anticipate that similar differences would be expected in female mice. Chronic erythroid expansion in β-thalassemia is associated with a thinning of cortical bone resulting in bone loss (*Haidar et al., 2011*; *Vogiatzi et al., 2006*). It is therefore surprising that patients with the more severe forms of β-thalassemia, namely TDT, in whom RBC transfusions lead to suppression of endogenous erythropoiesis, exhibit significantly greater decrements in BMD than NTDT patients (*Vogiatzi et al., 2010*). We have previously shown that ERFE is suppressed post-transfusion in TDT patients, and is significantly higher in NTDT patients (*Ganz et al., 2017*). Our finding of a marked reduction of bone mass in $Hbb^{th3/+}$ mice with genetically deleted *Erfe* (or $Hbb^{th3/+};Erfe^{-/-}$ mice) compared with $Hbb^{th3/+}$ mice provides strong evidence for a protective function of ERFE in preventing further worsening of the bone loss phenotype in β-thalassemia.

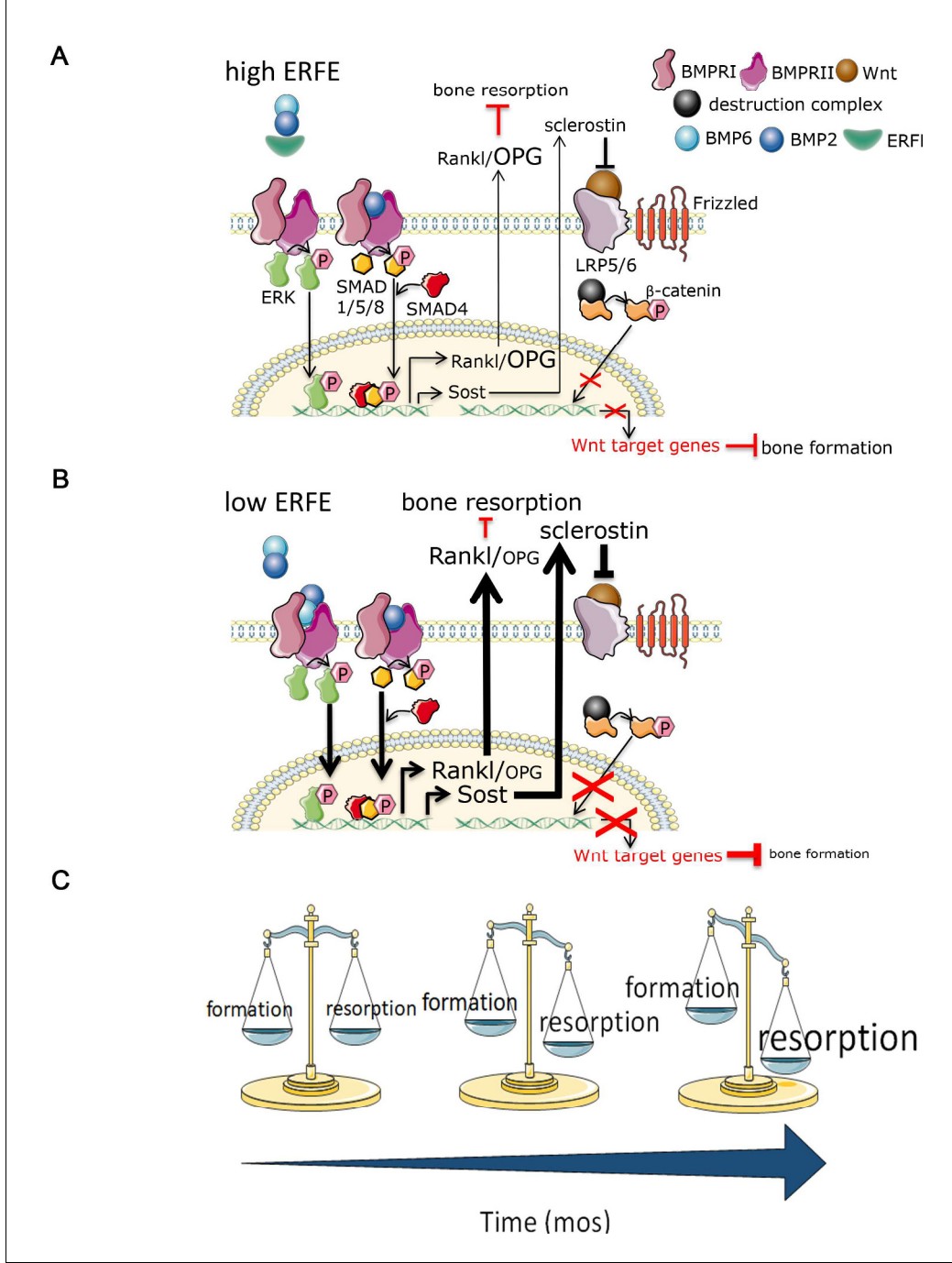

**Figure 6.** Putative osteoprotective function of ERFE in health and in β-thalassemia. In conditions of elevated ERFE (**A**), sich as β-thalassemia, more BMP2 and BMP6 is sequestered, decreasing signaling through the BMP/Smad and ERK pathways. This would result in decreased *Sost* and *Rankl* expression to decrease osteoclastogenesis and bone resorption. In contrast, when ERFE is low (**B**), increased BMP2, and possibly BMP6, leads to increased osteoclastogenesis with consequent decrease in bone formation. (**C**) Together, ERFE loss leads to a greater degree of progressively increased bone resorption relative to bone formation with age. ERFE = erythroferrone; BMP = bone morphogenetic protein; BMPR = BMP receptor; SOST = Sclerostin; RANKL = receptor activator of nuclear factor kappa-B ligand; OPG = osteoprotegrin; LRP = lipoprotein receptor-related protein; Wnt = wingless type MMTV integration site family.

Taken together, our findings uncover ERFE as a novel regulator of bone mass via its modulation of BMP signaling in osteoblasts. In addition, because RBC transfusion suppresses erythropoiesis and thus decreases ERFE in both mice (*Kautz et al., 2015*) and patients (*Ganz et al., 2017*) with β-thalassemia, a relative decrement of ERFE may explain the more severe bone disease in TDT than in NTDT patients. As a consequence, our findings identify ERFE as a promising new therapeutic target for hematologic diseases associated with bone loss, such as β-thalassemia.

## Acknowledgements

We sincerely appreciate Tomas Ganz and Chun-Ling ('Grace') Jung (UCLA), as well as Martina Rauner (University of Dresden) for many stimulating and helpful discussions and Ronald Hoffman (Icahn School of Medicine at Mount Sinai) for continued mentoring and advocacy. YZG acknowledges the support of the National Institute of Diabetes and Digestive and Kidney Diseases (NIDDK) (R01 DK107670 to YZG and DK095112 to RF, SR, and YZG). MZ acknowledges the support of the National Institute on Aging (U19 AG60917) and NIDDK (R01 DK113627). TY acknowledges the support of the National Institute on Aging (R01 AG71870). SR acknowledges the support of NIDDK (R01 DK090554) and Commonwealth Universal Research Enhancement (CURE) Program Pennsylvania.

## Additional information

### Competing interests

Vaughn Ostland, Huiling Han: is affiliated with Intrinsic Lifesciences, LLC. The author has no other competing interests to declare. Mone Zaidi: Deputy editor, *eLife*. The other authors declare that no competing interests exist.

### Funding

| Funder | Grant reference number | Author |
| --- | --- | --- |
| National Institute of Diabetes and Digestive and Kidney Diseases | DK107670 | Yelena Ginzburg |
| National Institute of Diabetes and Digestive and Kidney Diseases | DK095112 | Robert Fleming Stefano Rivella Yelena Ginzburg |
| National Institute of Diabetes and Digestive and Kidney Diseases | DK113627 | Mone Zaidi |
| National Institute on Aging | AG60917 | Mone Zaidi |
| National Institute of Diabetes and Digestive and Kidney Diseases | DK09055 | Stefano Rivella |
| National Institute on Aging | AG71870 | Tony Yuen |
| National Institute of Diabetes and Digestive and Kidney Diseases | DK090554 | Stefano Rivella |

The funders had no role in study design, data collection and interpretation, or the decision to submit the work for publication.

### Author contributions

Melanie Castro-Mollo, Data curation, Formal analysis, Investigation, Methodology, Writing - original draft, Writing - review and editing; Sakshi Gera, Data curation, Formal analysis, Validation, Investigation, Methodology, Writing - review and editing; Marc Ruiz-Martinez, Anisa Gumerova, Huiling Han, Data curation, Formal analysis, Investigation, Methodology, Writing - review and editing; Maria Feola, Data curation, Validation, Investigation, Methodology, Project administration, Writing - review

and editing; Marina Planoutene, Cara Clementelli, Data curation, Formal analysis, Writing - review and editing; Veena Sangkhae, Carla Casu, Resources, Data curation, Formal analysis, Investigation, Methodology, Writing - review and editing; Se-Min Kim, Data curation, Formal analysis, Methodology, Writing - review and editing; Vaughn Ostland, Resources, Data curation, Formal analysis, Investigation, Writing - review and editing; Elizabeta Nemeth, Resources, Data curation, Formal analysis, Supervision, Funding acquisition, Investigation, Methodology, Writing - review and editing; Robert Fleming, Stefano Rivella, Resources, Funding acquisition, Writing - review and editing; Daria Lizneva, Data curation, Formal analysis, Supervision, Investigation, Methodology, Writing - review and editing; Tony Yuen, Resources, Data curation, Formal analysis, Supervision, Investigation, Methodology, Writing - original draft, Writing - review and editing; Mone Zaidi, Conceptualization, Resources, Data curation, Supervision, Funding acquisition, Visualization, Methodology, Writing - original draft, Writing - review and editing; Yelena Ginzburg, Conceptualization, Resources, Data curation, Formal analysis, Supervision, Funding acquisition, Writing - original draft, Project administration, Writing - review and editing

## Author ORCIDs
Sakshi Gera (iD) http://orcid.org/0000-0002-1615-6259
Yelena Ginzburg (iD) https://orcid.org/0000-0002-3496-3783

## Ethics
Animal experimentation: This study was performed in strict accordance with the recommendations in the Guide for the Care and Use of Laboratory Animals of the National Institutes of Health. All of the animals were handled according to approved institutional animal care and use committee (IACUC) protocols (#16-0143) of the Icahn School of Medicine.

## Decision letter and Author response
Decision letter https://doi.org/10.7554/eLife.68217.sa1
Author response https://doi.org/10.7554/eLife.68217.sa2

## Additional files

### Supplementary files
• Source data 1. Source data file for presented studies.
• Transparent reporting form

### Data availability
All data generated or analysed during this study are included in the manuscript and supporting files.

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
