## [Decision Letter]

**Acceptance summary:**

The demonstration of an osteoprotective effect of erythroblast erythroferrone throughmodulating both osteoblastic and osteoclastic activity is novel and applicable not only to b-thalassemia but potentially to other conditions as well.

**Decision letter after peer review:**

Thank you for submitting your article "The Hepcidin Regulator Erythroferrone is a New Member of the Erythropoiesis-Iron-Bone Circuitry" for consideration by *eLife*. Your article has been reviewed by 3 peer reviewers, including Subburaman Mohan as the Reviewing Editor and Reviewer #2, and the evaluation has been overseen by Carlos Isales as the Senior Editor.

Essential revisions:

1. It appears that all of the experimental work was done in male mice. No rationale was provided for why only male mice were used. In this regard, a recent study showed that erythroid promoting effects of androgens are mediated via increasing erythropoietin expression and its downstream target, EFRE (McManus et al., Eur J Hematology, April 2020, PMID: 32311143). The authors should explain the rationale for choosing male mice in the context of published data and whether the findings are expected to be relevant in female mice.

2. Figure 1A shows that while L4-6 BMD was decreased, BV/TV was increased. It is unclear if the BV/TV was measured by histology. The authors should address the discrepancy between the decreased BMD versus increased BV/TV in the 6-week old mice.

3. While both BMP2 and BMP6 are known to interact with ERFE, the authors have focused on BMP2. Did the authors measure BMP6 levels in the conditioned medium of EFRE knockout osteoblasts? Do the authors believe that BMP2 is the major mediator of ERFE effects in osteoblasts to regulate osteoclast functions?

4. Mice lacking ERFE show increased bone formation rates at younger age despite increased expression of Sclerostin. It would help the authors more extensively discuss those findings, and modify the schematic figure accordingly.

5. n 5-month-old Erfe-/-mice, the result showed enhanced osteoblast differentiation and mineralization with enhanced expression of Runx2and Sp7, and Sost and Tnfsf11 in Figure 4A and B, however, in Figure 1D, there is no differences in MS, MAR, and BFR.

6. In Figure 3C, it is not clear why Smad1 and ERK 1/2 level were changed in ERFE deleted cells and wild type cells when treated with or without BMP2, while pSmad1/5/8 did not respond to BMP2 stimulation in the wild type group, which are inconsistent to published results in osteoblasts.

7. To probe the mechanism of action of ERFE on osteoblastic bone formation and osteoclastic bone resorption, bone marrow cells were induced with osteoblastic differentiating media for different times. It is likely that the cultures contain other cell types besides osteoblast line cells at early time points.

8. The authors should discuss potential explanation for the differences in skeletal phenotype between ERFE knockout and control mice at 6-week-old vs 5-month-old animals.

In this study, Castro-Mollo and colleagues report the novel and groundbreaking finding that erytroferrone (ERFE) is produced and secreted by osteoblasts, i.e. the cells forming bone, at high levels and in an EPO-independent fashion. More importantly, ERFE regulates bone mass as its loss leads to a low bone mass phenotype.. The bone loss is secondary to an increase of bone resorption due to an augmented production of RANKL and sclerostin by osteoblasts via higher availability of BMPs.

The paper provides novel and exciting information, which is likely to open a new field of investigation. The study, which is both highly mechanistic and translationally relevant, spurs a wealth of interesting questions. The findings are solid and convincing from both a biological and a translational standpoint. The authors' conclusions are supported by the data as shown.

The study is highly significant for the treatment of low bone mass in patients with β-thalassemia, which is a yet unexplored field of research. Β-Thalassemia is a devastating genetic disease in which severe anemia is associated with high levels of ERFE. While the data in general support the overall conclusion regarding a role for EFRE in the regulation of bone metabolism, the authors should address the recommended revisions of the review panel mentioned above to improve the clarity of the manuscript.

---

## [Author Response]

Essential revisions:1. It appears that all of the experimental work was done in male mice. No rationale was provided for why only male mice were used. In this regard, a recent study showed that erythroid promoting effects of androgens are mediated via increasing erythropoietin expression and its downstream target, EFRE (McManus et al., Eur J Hematology, April 2020, PMID: 32311143). The authors should explain the rationale for choosing male mice in the context of published data and whether the findings are expected to be relevant in female mice.

We anticipated that the gender effect on erythropoiesis and bone metabolism would require separate evaluation of the genders and selected male mice for efficient use of our mouse colony. We anticipate that the effects of ERFE loss would also yield significant differences in female mice. Ultimately, a separate evaluation in female mice is also warranted. McManus et al. note that androgens promote erythropoiesis by a DNA binding-dependent mechanism to stimulate Epo expression in non-hematopoietic cells, leading to ERFE production by erythroblasts. Importantly, McManus et al. demonstrate no differences in serum Epo or bone marrow *Erfe* expression between male and female wild type mice (vehicle treated). As a consequence, we anticipate that differences between female wild type and *Erfe*^-/-^ as well as between *Hbb^th3/+^* and *Hbb^th3/+^*;*Erfe*^-/-^ mice would reflect differences similar to those presented in male mice in the current manuscript. Finally, given the greater bone mineral density in male relative to female C57BL/6 mice [Glatt J Bone Miner Res 2007], ERFE loss may have even more pronounced effects in female relative to male mice.

We have added the following statement to the Discussion section (page 15):

“The results presented reflect evidence derived from analysis of male mice. We anticipate that similar differences would be expected in female mice.”

2. Figure 1A shows that while L4-6 BMD was decreased, BV/TV was increased. It is unclear if the BV/TV was measured by histology. The authors should address the discrepancy between the decreased BMD versus increased BV/TV in the 6-week old mice.

We appreciate this request for clarification. BV/TV was measured using Von Kossa stained histomorphometry slides; this has been noted in the Skeletal Phenotyping part of the Methods section. Furthermore, we hypothesize that ERFE loss directly enhances osteoblastogenesis, leading to increased BV/TV, and directly enhances osteoclastogenesis, ultimately leading to decreased bone mineral density (see also response to point #4). We hypothesize that the increase in BV/TV is transient and disappears as the mice age, when the cumulative effect on bone resorption is evident. We have added a new panel to Figure 6, namely Figure 6C, to specifically provide a visual representation of this point.

3. While both BMP2 and BMP6 are known to interact with ERFE, the authors have focused on BMP2. Did the authors measure BMP6 levels in the conditioned medium of EFRE knockout osteoblasts? Do the authors believe that BMP2 is the major mediator of ERFE effects in osteoblasts to regulate osteoclast functions?

We appreciate this query. Taken together, our findings that serum BMP2 concentration is elevated in *Erfe^-/-^* mice, consistent with increased BMP2 concentration in cultured *Erfe^-/-^* osteoblast supernatants, and the diminished effect of added BMP2 on signaling in *Erfe^-/-^* relative to WT osteoblasts, provide definitive in vivo and in vitro evidence of ERFE-dependent effects on BMP2.

As the reviewer states, while ERFE has been found to sequester both BMP6 and BMP2 [Arezes Blood 2018; Wang Blood 2020], we focused on BMP2 given its role in bone remodeling [Salazar Nat Rev Endocrinol 2016]. No reagents are available for easily measuring BMP6 in osteoblast conditioned media. To test whether an ERFE effect on bone is BMP2 specific, we evaluated the effect of BMP6 in osteoblasts in vitro, demonstrating similar responses in signaling to those observed with BMP2 in wild type and *Erfe*^-/-^ osteoblasts in vitro (Figure 3C and 3D). We have added this data to new Figure 3—figure supplement 3 in the revised manuscript.

Finally, in response to the reviewer’s question, we believe that increased BMPs lead to activation of BMP signaling in *Erfe*^-/-^ osteoblasts to stimulate osteoblast differentiation. This is based on the observation that BMPs are increased in the supernatant of *Erfe*^-/-^ osteoblasts leading to greater activation of BMP signaling and a blunted effect in response to exogenously added BMPs. However, specifically which BMP is the primary physiological actor in this pathway is incompletely understood. To address this, we have edited this part of the Results section (page 12) and the Discussion section to reflect an ERFE effect on bone via sequestration of BMPs more generally.

4. Mice lacking ERFE show increased bone formation rates at younger age despite increased expression of Sclerostin. It would help the authors more extensively discuss those findings, and modify the schematic figure accordingly.

We appreciate this request for clarification. The reviewer correctly notes that while the in vivo data demonstrates increased bone formation rate in *Erfe*^-/-^ relative to wild type mice (Figure 1C), in vitro data demonstrates increased *Sost* expression in cultured osteoblasts from *Erfe*^-/-^ relative to wild type mice (Figure 4B) which would be expected to decrease—rather than increase—the bone formation rate. It has been shown that sclerostin inhibits Wnt signaling, providing negative feedback and preventing further osteoblast differentiation [Tanaka J Bone Miner Metab 2021]. Sclerostin also optimizes the relative proportion of RANKL and OPG to induce bone resorption [Takayanagi Nat Rev Immun 2007]. Our results confirm decreased *Opg* expression (Figure 4B) and increased RANKL concentration (Figure 4C) in *Erfe*^-/-^ relative to wild type osteoblasts in vitro. Taken together, these findings strongly suggest that while ERFE loss leads to increased bone mineralization in vitro, in vivo effects of enhanced osteoblast function and gene expression result in enhanced osteoclastogenesis in *Erfe*^-/-^ relative to wild type mice (Figure 1F), culminating in decreased bone mineral density.

Furthermore, we have added additional statistical calibration to Figure 1 to provide evidence that the phenotype in younger mice is similar to that in older mice and have edited the Results section accordingly.

Finally, in response to this query, we have expanded Figure 6 to include Figure 6C to visually address this point and added the following paragraph to the Discussion section (page 15):

“Bone is a highly dynamic and purposefully organized tissue which undergoes constant remodeling in response to changing metabolic and mechanical needs. […] Our findings demonstrate that while ERFE loss leads to increased bone mineralization in vitro and bone formation increases as expected with age, the composite effect in vivo results in greater enhancement of osteoclastogenesis relative to osteoblastogenesis in *Erfe*^-/-^ relative to wild type mice, consequently decreasing bone mineral density (Figure 6C).”

5. n 5-month-old Erfe-/-mice, the result showed enhanced osteoblast differentiation and mineralization with enhanced expression of Runx2and Sp7, and Sost and Tnfsf11 in Figure 4A and B, however, in Figure 1D, there is no differences in MS, MAR, and BFR.

We appreciate the differences between in vivo and in vitro results and interpret the query as a request for comment on this point. It is our interpretation that while in vitro results enable cell-autonomous effect of ERFE loss in osteoblasts, the effects in vivo are influenced also by the osteoblast-dependent effects on osteoclasts (see also response to #2 and #4). As a consequence, increased bone mineralization in vitro in *Erfe*^-/-^ relative to wild type osteoblasts results in increased *Tnfsf11* and decreased *Opg* mRNA expression as well as increased RANKL concentration in the supernatant. These findings would be expected to cause increased osteoclast surface and osteoclast number in vivo as we demonstrate in 5-month-old *Erfe*^-/-^ relative to wild type mice to explain a lack of difference in MS, MAR, and BFR between *Erfe*^-/-^ and wild type mice.

6. In Figure 3C, it is not clear why Smad1 and ERK 1/2 level were changed in ERFE deleted cells and wild type cells when treated with or without BMP2, while pSmad1/5/8 did not respond to BMP2 stimulation in the wild type group, which are inconsistent to published results in osteoblasts.

We regret not making this more clear and have re-organized Figure 3C and 3D for further clarity. pSmad1/5/8 and pERK1/2 both increase in *Erfe*^-/-^ relative to wild type osteoblasts. Furthermore, pSmad1/5/8 and pERK1/2 are both increased in response to BMP2 in wild type but not *Erfe*^-/-^ osteoblasts. No differences are observed in pp38/MAPK signaling.

7. To probe the mechanism of action of ERFE on osteoblastic bone formation and osteoclastic bone resorption, bone marrow cells were induced with osteoblastic differentiating media for different times. It is likely that the cultures contain other cell types besides osteoblast line cells at early time points.

We appreciate this potential confounder. However, a hierarchical differentiation tree available from BloodSpot demonstrates the erythroid specificity (MEP) of ERFE (gene name Fam132b) expression in healthy individuals:

**Author response image 1. sa1fig1:** 

Furthermore, our data demonstrates that alkaline phosphatase, a well-established marker of osteoblastic lineage, is increased as early as day 3 in culture (new Figure 2—figure supplement 2), confirming the presence of osteoblasts. Taken together, the lack of Epo in these culture conditions coupled with the lack of adhesiveness of erythroid lineage cells and rapid increase in alkaline phosphatase together strongly support the osteoblast-specific production of ERFE in this culture system. This information has been added to the Methods and Results sections.

8. The authors should discuss potential explanation for the differences in skeletal phenotype between ERFE knockout and control mice at 6-week-old vs 5-month-old animals.

We appreciate this request for clarification. We intentionally compared growing (6-week-old) and mature (5-month-old) mice to assess the potentially distinct or cumulative effect of ERFE loss on bone growth and / or remodeling, respectively. Our results demonstrate that ERFE loss leads to impaired BMD in both cohorts of mice. However, while MS/BS, MAR, BFR and BV/TV are increased in 6-week-old *Erfe*^-/-^ relative to wild type mice, no differences are evident between 5-month-old *Erfe*^-/-^ and wild type mice. These findings strongly suggest that over time, while both bone formation and resorption increase relative to younger mice, the balance between them favors bone resorption (new Figure 6C). This interpretation is further corroborated by a more profound increase in osteoclast surface and osteoclast number in 5-month-old *Erfe*^-/-^ relative to wild type than in 6-week-old mice.

We have added the following statement to the Discussion section (page 15) to address this point:

”We intentionally compared growing (6-week-old) and mature (5-month-old) mice to assess the potentially distinct or cumulative effect of ERFE loss on bone growth and / or remodeling, respectively. […] We are enthusiastic about the merits of this work and confident that it will advance the exploration and potential application of ERFE-related therapeutics in patients with disordered erythropoiesis and iron metabolism.